

# Diagnosing spatial error structures in CO$_2$ mole fractions and XCO$_2$ column mole fractions from atmospheric transport

Thomas Lauvaux[1,a], Liza I. Díaz-Isaac[1,b], Marc Bocquet[2], and Nicolas Bousserez[3,c]

[1]Department of Meteorology and Atmospheric Science, The Pennsylvania State University, University Park, PA
[2]CEREA, joint laboratory École des Ponts ParisTech and EDF R& D, Université Paris-Est, Champs-sur-Marne, France
[3]University of Colorado Boulder, Boulder, CO
[a]Now at Laboratoire des Sciences du Climat et de l'Environnement, CEA, CNRS, UVSQ/IPSL, Université Paris-Saclay, Orme des Merisiers, 91191 Gif-sur-Yvette cedex
[b]Now at Scripps Institution of Oceanography, University of California, San Diego, 92093, USA
[c]Now at European Centre for Medium-Range Weather Forecasts, Reading, UK

*Correspondence to:* Thomas Lauvaux (tul5@psu.edu)

**Abstract.** Atmospheric inversions inform about the magnitude and variations of greenhouse gas (GHG) sources and sinks from global to local scales. Deployment of observing systems such as spaceborne sensors and ground-based instruments distributed around the globe has started to offer an unprecedented amount of information to estimate surface exchanges of GHG at finer spatial and temporal scales. However, inversion methods still rely on imperfect atmospheric transport models of which error

structures directly affect the inverse estimates of GHG fluxes. The impact of spatial error structures on the inverse fluxes increase concurrently with the density of the available measurements. In this study, we diagnose the spatial structures due to transport model errors affecting modeled in situ carbon dioxide (CO$_2$) mole fractions and total column dry air mole fractions of CO$_2$ (XCO$_2$). We implemented a cost-effective filtering technique recently developed in the meteorological data assimilation community to describe spatial error structures using a small-size ensemble. This technique can enable ensemble-based error

analysis for multi-year inversions of sources and sinks. The removal of noisy structures in our small-size ensembles is evaluated by comparison to larger-size ensembles. A second filtering approach for error covariances is proposed (Wiener filter), producing similar results over the 1-month simulation period than a Schur filter. We conclude that key information about error variances and spatial error correlation structures are recoverable from small-size ensembles of about ten (10) members down to five (5), improving the representation of transport errors in mesoscale inversions of CO$_2$ fluxes. Moreover, error variances of in

situ near-surface and free-tropospheric CO$_2$ mole fractions differ significantly from total column XCO$_2$ error variances. We conclude that error variances for remote sensing observations need to be quantified independently of in situ CO$_2$ mole fractions due to the complexity of spatial error structures at different altitudes. However, we show the potential use of meteorological error structures such as the mean horizontal wind speed, directly available from Ensemble Prediction Systems, to approximate spatial error correlations of in situ CO$_2$ mole fractions, with similarities in seasonal variations and characteristic error length

scales.





## 1  Introduction

Atmospheric carbon dioxide ($CO_2$) mole fraction has been increasing steadily since the first industrial revolution, primarily due to fossil fuel emissions and land use change (Ciais et al., 2015). Recent estimates of sources and sinks at the global scale suggest a coincidental reinforcement of natural sinks balancing the continuously increasing anthropogenic emissions (Le Quéré et al., 2016; Keenan et al., 2016). Therefore, the fraction of fossil fuel $CO_2$ remaining in the atmosphere was kept constant at 2ppm per year[1], excluding short-time anomalies such as El Niño events (Feely et al., 1999; Kim et al., 2016). In the objective of characterizing the natural sink mechanisms, atmospheric inversion methods have provided some evidences of a fertilization effect possibly increasing the effective absorption by plants of the exceeding $CO_2$ in the atmosphere (Schimel et al., 2015). But large uncertainties still affect atmospheric inversions of $CO_2$ fluxes and limit the interpretation of continental-scale $CO_2$ budgets (Peylin et al., 2013). Therefore, more robustness in these findings first requires better error characterization affecting inverse estimates (Baker et al., 2007; Stephens et al., 2007; Díaz-Isaac et al., 2014).

Atmospheric inversions of Greenhouse Gases (GHG) are now widely used to infer surface fluxes from natural (e.g. Enting, 2002; Gurney et al., 2002; Lauvaux et al., 2012; Peylin et al., 2013) and anthropogenic (e.g. McKain et al., 2015; Lauvaux et al., 2016) sources at global, regional, and local scales. However, key information in carbon cycle science lies in multi-year time scales, therefore confining the development of inverse methodologies to cost-effective approaches (e.g. Bruhwiler et al., 2005). Based on similar methodologies than meteorology or geophysics, atmospheric inversions have used primarily fast approaches to produce multi-decadal fluxes such as variational approaches (Baker et al., 2006; Chevallier et al., 2010), avoiding large ensemble of simulations based on Monte Carlo formulation (Evensen, 1994). In parallel, assumptions made in prior flux errors and transport errors impact the inverse solution in similar ways (Engelen et al., 2002). Concerning the prior flux errors, few studies have proposed to constrain the spatial and temporal structures more rigorously (Wu et al., 2013; Ganesan et al., 2014), some of them based on terrestrial biogeochemical models and eddy-covariance flux measurements to estimate the spatial structures in prior flux errors of $CO_2$ (e.g. Chevallier et al., 2006; Hilton et al., 2013). For transport errors, correlations remained small at the global scale, primarily due to sparse atmospheric GHG observation networks. However, denser tower networks (Andrews et al., 2014) and recent satellite missions have significantly increased the sampling density (e.g. the Greenhouse gases Observing SATellite (GOSAT; Yokota et al. (2009); Houweling et al. (2015)) and the Orbital Carbon Observatory (OCO-2) missions (Crisp et al., 2004)) requiring the characterization of their correlated errors in inversion systems.

The increased density in existing tower networks and the availability of fine-scale satellite retrievals raised concerns about spatial and temporal structures in transport model errors (Rayner and O'Brien, 2001; Lauvaux et al., 2009; Miller et al., 2015). The proximity of the measurements (e.g. couple kilometers between OCO-2 retrievals) means that spatial correlations in model errors are significant and can no longer be ignored (Chevallier, 2007). This issue becomes critical to greenhouse gas inversion problems when applied to urban scales (Lauvaux et al., 2016) but remains poorly studied to date. Recent deployment of path-integrated instruments also increased the complexity of the problem from the ground when trying to invert for emissions from single facilities such as a large dairy (Viatte et al., 2017).

---

[1]www.esrl.noaa.gov/gmd/ccgg/trends/





Ensemble approaches are useful to describe flow-dependent errors (e.g. Anderson, 2001; Evensen, 2003) but remain computationally expensive due to the number of model simulations required to correctly represent model error statistics (Houtekamer and Mitchell, 1998). In general, a small number of members leads to incomplete descriptions of error structures which require the use of localization to avoid spurious correlations (Houtekamer and Mitchell, 2001; Raynaud and Pannekoucke, 2013). But

small-size ensembles are efficient computationally and able to provide information on flow-dependent error structures compared to prescribed static error structures (Brousseau et al., 2012). With the development of new perturbation methods, the number of members may decrease significantly thanks to optimal perturbations combining physics, parameter sensitivity and energy-based perturbations (Jankov et al., 2017). In any case, small-size ensembles remain affected by sampling noise which has to be removed before extracting spatial structures, either by modeling (Pannekoucke et al., 2008; Lauvaux et al., 2009) or

by filtering unphysical structures (Hamill et al., 2001; Houtekamer and Mitchell, 2001). Here, we apply a newly developed approach based on local filtering and a localization technique (Ménétrier et al., 2015a; Ménétrier et al., 2015b). There are only a few approaches for the optimal localization of covariance matrices in the field of data assimilation for the geosciences (Lei and Anderson, 2014; Flowerdew, 2015; De La Chevrotière and Harlim, 2017). To our knowledge, the method is the only one so far which is both (i) mathematically consistent and (ii) a priori, i.e. not based on learning on past or present datasets. Besides,

in spite of its sophistication, the filtering approach is rather straightforward to implement.

In this study, we apply the filter of variances and the covariance localization developed in Ménétrier et al. (2015a), and propose an additional optimal solution to using the optimality condition, both for Gaussian and non-Gaussian error statistics cases. The filter is applied to several calibrated ensembles of different sizes to evaluate the impact of our filter on small (5 members) to larger (25 members) to the full ensemble (45 members) based on multi-physics simulations (Díaz-Isaac et al., 2018a). Results

are presented for in situ $CO_2$ mole fractions, $XCO_2$ dry air mole fractions, mean horizontal winds, and Planetary Boundary Layer heights (PBLH). We discuss the results in Section 4.

## 2 Methods

### 2.1 Calibration of WRF-CO$_2$ ensembles

We generate an ensemble using the Weather Research and Forecasting (WRF) model version 3.5.1 (Skamarock et al., 2008),

including the chemistry module modified in this study for $CO_2$ (Lauvaux et al., 2012). The ensemble consists of 45 members that were generated by varying the different physics parameterization and meteorological data. The land surface models, surface layers, planetary boundary layer schemes, cumulus schemes, microphysics schemes, and meteorological data (i.e., initial and boundary conditions) are alternated in the ensemble (Díaz-Isaac et al., 2018b). All the simulations use the same radiation schemes, both long and shortwave. The different simulations were run using the one-way nesting method, with two

nested domains. The coarse domain uses a horizontal grid spacing of 30km and covers most of the United States and part of Canada. The inner domain uses a 10km grid spacing, is centered in Iowa and covers the Midwest region of the United States. The vertical resolution of the model is described with 59 vertical levels, with 40 of them within the first 2km of the atmosphere. This work focuses on the WRF simulation with the higher resolution, therefore only the 10-km domain will be



analyzed. The simulation was performed from 27 June, 2008 to 21 July 2008, with a 10-day spin-up for initial conditions. The $CO_2$ fluxes for summer 2008 were obtained from NOAA Global Monitoring Division's CarbonTracker version 2009 (CT2009) data assimilation system (Peters et al. (2007), with updates documented at http://carbontracker.noaa.gov). The different fluxes that CT2009 propagates into the models are fossil fuel burning, terrestrial biosphere exchange, and the exchange with oceans.

The $CO_2$ lateral boundary conditions were obtained from CT2009 mole fractions. Only the atmospheric transport fields vary between each model configuration or ensemble member.

The ensemble was calibrated over the Midwest U.S. using the available meteorological observations and the 10-km model simulation as described in Díaz-Isaac et al. (2018a). The measurements used included balloon soundings collected over the Midwest region (http://weather.uwyo.edu/upperair/sounding.html) for 14 rawinsonde stations. The ensemble was calibrated for

three different meteorological variables: wind speed, wind direction and planetary boundary layer height (PBLH) in the late afternoon data (i.e., 0000 UTC) from the different rawinsondes. Daytime data was used to represent well mixed conditions, at the selected time when $CO_2$ mole fraction are assimilated in atmospheric inversions to avoid stable conditions near the surface. The calibration process is described in Garaud and Mallet (2011), selecting optimal ensembles of different sizes using Simulated Annealing and Genetic Algorithm techniques. The metric used in Díaz-Isaac et al. (2018a) is the flatness

of the Rank Histogram which measures the dispersion of an ensemble. By eliminating members with redundant information, smaller ensembles were able to better match the variability in the observations. We refer to Díaz-Isaac et al. (2018a) for a full description of the calibration process and the final selection of optimal ensembles.

Here, we will compare the different ensembles generated in Díaz-Isaac et al. (2018a) from 5 to 8 to 10 members. An additional ensemble was created for our study with a larger number of members in order to address the potential lack of

representativeness of model errors with small-size ensembles. Therefore, we generated a 25-member ensemble and applied the same calibration process. This ensemble has not been documented in Díaz-Isaac et al. (2018a) but is described here in the Appendix A. We compare the results of the filtering for small sizes to the 25-member calibrated ensemble instead of the original 45-member ensemble that was not calibrated.

## 2.2   Variance Filtering and Covariance Localization

Ménétrier et al. (2015a); Ménétrier et al. (2015b) have proposed a new theory for the optimal filtering of sample variances and covariances. These are defined by the following empirical second-order moment statistics. Assume we have an ensemble of states $\mathbf{x}_k \in \mathbb{R}^n$ for $k = 1, \ldots, N$ of mean $\overline{\mathbf{x}}$, from which to infer the statistics. Define the associated anomalies $\delta\mathbf{x}^k = \mathbf{x}^k - \overline{\mathbf{x}}$, also called perturbations. Then, the sample covariance matrix is:

$$\widetilde{\mathbf{B}} = \frac{1}{N-1} \sum_{k=1}^{N} \delta\mathbf{x}^k \left(\delta\mathbf{x}^k\right)^{\mathrm{T}}, \tag{1}$$

which is an unbiased estimator of the true covariance matrix $\mathbf{B}^\star$, i.e., $\mathbb{E}\left[\widetilde{\mathbf{B}}\right] = \mathbf{B}^\star$ where $\mathbb{E}$ is the expectation operator over the reference distribution from which the $\mathbf{x}_k$ are sampled. In the following, we denote by $\widetilde{B}_{ij}$ the entries of this covariance matrix.





Filtering of variances and covariances is made necessary because of the finite-size of the sample ensembles, which can generate significant sampling errors. The sampling errors can be filtered out by applying a linear filter on the variances and covariances. The most general linear filter is of the form

$$\widehat{B}_{ij} = \sum_{kl} F_{ijkl} \widetilde{B}_{kl}, \tag{2}$$

where $\widehat{\mathbf{B}}$ is the filtered error covariance matrix. Typical examples are the application of a convolution to the vector of variances to smooth them out, or the application of a Schur product with a non-degenerate, short-range correlation matrix to the sample covariance matrix. The linear filter often requires parameters, a correlation length typically, that must be tuned for the filter to be optimal.

The theory proposed in Ménétrier et al. (2015a) to achieve optimality of the filter is based on three key ingredients:

1. The first one consists in requiring that the residual sampling error be minimal. Assume that we have an estimator $\widetilde{\mathbf{x}}$ of some statistics of a reference distribution with true statistics $\mathbf{x}^\star$, obtained from sampling from this distribution. We regularize $\widetilde{\mathbf{x}}$ with a linear filter $\mathbf{F}$ (a matrix here) in order to minimize the sampling error: $\widehat{\mathbf{x}} = \mathbf{F}\widetilde{\mathbf{x}}$. A typical criterion to minimize is

$$\mathcal{L}(\mathbf{F}) = \mathbb{E}\left[ (\mathbf{x}^\star - \mathbf{F}\widetilde{\mathbf{x}})(\mathbf{x}^\star - \mathbf{F}\widetilde{\mathbf{x}})^{\mathrm{T}} \right]. \tag{3}$$

The variation of this criterion with respect to a variation $\delta\mathbf{F}$ is $\delta\mathcal{L}(\mathbf{F}) = -2\mathbb{E}\left[ (\mathbf{x}^\star - \mathbf{F}\widetilde{\mathbf{x}})\widetilde{\mathbf{x}}^{\mathrm{T}} \right] \delta\mathbf{F}^{\mathrm{T}}$, which implies, that, at the minimum, we have an optimality condition in the form of an orthogonality of random vectors:

$$\mathbb{E}\left[ (\mathbf{x}^\star - \mathbf{F}\widetilde{\mathbf{x}})\widetilde{\mathbf{x}}^{\mathrm{T}} \right] = \mathbb{E}\left[ (\mathbf{x}^\star - \widehat{\mathbf{x}})\widetilde{\mathbf{x}}^{\mathrm{T}} \right] = \mathbf{0}. \tag{4}$$

This is a linear equation in $\mathbf{F}$ whose solution is

$$\mathbf{F}^\star = \left\{ \mathbb{E}\left[ \widetilde{\mathbf{x}}\widetilde{\mathbf{x}}^{\mathrm{T}} \right] \right\}^{-1} \mathbb{E}\left[ \mathbf{x}^\star \widetilde{\mathbf{x}}^{\mathrm{T}} \right]. \tag{5}$$

If $\mathbf{F}$ is a Schur filter, i.e., $\widehat{\mathbf{x}} = \mathbf{f} \circ \widetilde{\mathbf{x}}$, given by the Schur or Hadamart product (which is a subcase of the above problem) – hence $\mathbf{F}$ is now a vector $\mathbf{f}$ – then the solution has the form

$$\mathbf{f}^\star = \frac{\mathbb{E}\left[ \mathbf{x}^\star \circ \widetilde{\mathbf{x}} \right]}{\mathbb{E}\left[ \widetilde{\mathbf{x}} \circ \widetilde{\mathbf{x}} \right]}, \tag{6}$$

where the division of vectors is component-wise. Equations (5,6) can be applied to the filtering of $\widetilde{\mathbf{B}}$, storing the entries $B_{ij}$ in $\mathbf{x}$. Hence, they provide optimality conditions for linear filtering of $\widetilde{\mathbf{B}}$. They are known in the signal scientific
community as Wiener filters.

2. The second one is to exploit the structure relationships that bind the moments of sample estimators of the reference distribution. For any reference distribution (referred to as the non-Gaussian case in the following), the second-order moments of the sample covariances $B_{ij}$ are functions of the second-order and fourth-order moments of the reference


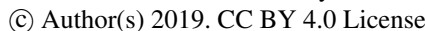


distribution. If, in addition, one assumes this reference distribution to be a Gaussian, then the covariances of the sample covariances $\widetilde{B}_{ij}$ are only functions of the second-order moments of the reference distribution. This will be naturally referred to later as the Gaussian case. For instance, in the Gaussian case, the relation has the well-known form:

$$\mathbb{E}\left[\left(B_{ij}^{\star} - \widetilde{B}_{ij}\right)^2\right] = \frac{1}{N-1}\left(B_{ij}^{\star}B_{ij}^{\star} + B_{ii}^{\star}B_{jj}^{\star}\right). \tag{7}$$

5    3. In spite of the above key ideas, some local spatial averaging will additionally be needed to obtain robust estimators for the filters and their correlation lengths. Such averaging can be justified by ergodic assumptions on the statistics of the errors.

In the following, we make the difference between the cases where the true distribution is assumed Gaussian or not, since we saw it has an impact on the structure function such as (7), and could yield distinct optimal filtering results.

## 2.2.1 Gaussian case

It turns out that it is more convenient to filter the variance and the correlation independently, in particular using a general linear filter for the variances and a Schur filter for the correlation.

We denote $\mathbf{v}$ the vector of variances, i.e., $v_i \equiv B_{ii}$. Combining the optimality criterion (4) with the structure relationship (7), without reference to any explicit filter at this stage, the filtered and the sampled variances are related by (see Eq. (50) of Ménétrier et al., 2015a):

$$\mathfrak{C}_i^{\mathrm{G}} \equiv \mathbb{E}[\widetilde{v}_i^2] - \frac{N+1}{N-1}\mathbb{E}[\widetilde{v}_i\widehat{v}_i] = 0. \tag{8}$$

If we filter the covariances with a Schur filter, i.e., $\widehat{\mathbf{B}} = \mathbf{F} \circ \widetilde{\mathbf{B}}$, then one obtains (see Eq. (64) of Ménétrier et al., 2015a):

$$F_{ij}^{\mathrm{G}} = \frac{N-1}{(N+1)(N-2)}\left\{(N-1) - \frac{\mathbb{E}[\widetilde{v}_i\widetilde{v}_j]}{\mathbb{E}[\widetilde{B}_{ij}^2]}\right\}. \tag{9}$$

## 2.2.2 Non-Gaussian case

In the non-Gaussian case, the structure relationship incorporates a term that depends on the fourth-order moments $\Xi_{ijkl}$ of the true error statistics. Using these relationships and the optimality criterion (4), without reference at this stage to any particular filter, one obtains (see Eq. (48) of Ménétrier et al., 2015a):

$$\mathfrak{C}_i^{\mathrm{NG}} \equiv \mathbb{E}[\widetilde{v}_i^2] - \frac{N(N-2)(N-3)}{(N-1)(N^2-3N+3)}\mathbb{E}[\widetilde{v}_i\widehat{v}_i]$$
$$- \frac{N^2}{(N-1)(N^2-3N+3)}\mathbb{E}[\widetilde{\Xi}_{iiii}] = 0. \tag{10}$$





Again, but in the non-Gaussian case, if we regularize the covariances with a Schur filter, i.e., $\widehat{\mathbf{B}} = \mathbf{F} \circ \widetilde{\mathbf{B}}$, then ones obtains the optimal filter (see Eq. (62) of Ménétrier et al., 2015a):

$$
\begin{aligned}
F_{ij}^{\text{NG}} = & \frac{(N-1)^2}{N(N-3)} - \frac{N}{(N-2)(N-3)} \frac{\mathbb{E}[\widetilde{\Xi}_{ijij}]}{\mathbb{E}[\widetilde{B}_{ij}^2]} \\
& + \frac{N-1}{N(N-2)(N-3)} \frac{\mathbb{E}[\widetilde{v}_i \widetilde{v}_j]}{\mathbb{E}[\widetilde{B}_{ij}^2]}.
\end{aligned}
\tag{11}
$$

For the localization of the covariances and hence the correlations, Eqs. (9,11) provide the optimal Schur localization. For the filtering of the variances, one uses Eqs. (8,10) but still need to specify a filter, such as a convolution with a short-range kernel of correlation length $l$. Then Eqs. (8,10) are implicit equations for $l$, which can be solved iteratively using, for instance, a fixed point method.

We note that all these formulae still depend on some statistical expectation, such as $\mathbb{E}[\widetilde{B}_{ij}^2]$. To make those formulae practical,
we identify these expectations as local, if not global, spatial averages.

### 2.2.3   Wiener filter

There is an alternative to using the optimality condition (4) in conjunction with the structure relationships of the moments of $\widetilde{\mathbf{B}}$. We propose to solely use the optimality condition (4) and upon choosing the generic form of the filter use the optimal filters given by (5) or (6). We will call them Wiener filters in the following.

For instance, assuming Schur regularization, we obtain the Wiener filter:

$$
L_{ij} = \frac{\mathbb{E}[\widetilde{B}_{ij} B_{ij}^\star]}{\bar{\mathbb{E}}[\widetilde{B}_{ij}^2]}.
\tag{12}
$$

Using the sample estimator $B_{ij}^\star \approx \widetilde{B}_{ij}$, we obtain the approximation:

$$
L_{ij} \approx \frac{\mathbb{E}^2[\widetilde{B}_{ij}]}{\mathbb{E}[\widetilde{B}_{ij}^2]}.
\tag{13}
$$

Both Wiener and Schur filters will be applied to sub-domains defined around instrumented tower locations measuring con-
tinuously $CO_2$ mole fractions in the US Upper Midwest (Miles et al., 2012). The sub-domains cover an area of 400x400 km$^2$ around each site (here seven sites across the domain) which also correspond to the spatial extent of the local spatial averaging (3rd item in Section 2.2). Due to computational limitations, we performed additional experiments with larger sub-domains for our 25-member ensemble, as show in the Section 3.5.

### 2.3   Meteorology and $CO_2$ error structures

We want to explore the relationships between the different variables especially in situ mole fractions of $CO_2$, total column $XCO_2$, and the PBL height. We will compare both the error variances and covariances to identify possible links between error structures in PBL depth and $CO_2/XCO_2$ mole fractions. We will explore the spatial correlation lengths for $CO_2$ mole fractions,





mean horizontal wind (zonal and latitudinal components), and PBL depths to quantify and possible utilize error structures in meteorological fields to generate $CO_2$ and $XCO_2$ error structures. Most Ensemble Prediction Systems (EPS's) provide spatial error correlations for meteorological variables which could be used to construct error covariances for $CO_2$ and $XCO_2$. Error covariances of $CO_2$ mole fractions depend on the $CO_2$ fluxes, but error structures in the atmospheric models should remain

independent of the $CO_2$ flux distribution. Díaz-Isaac et al. (2018a) show that first-order discrepancies in PBL depth seem related to large errors in $CO_2$. Here, we investigate further the links between errors across different variables. We present the results in Section 3.4 for the variances and in section 4.3 for the error correlations.

## 3   Results

### 3.1   Sampling noise due to ensemble size

We computed the variances over the domain from the 5-, 8-, 10- and 25-member ensembles as shown in Figure 1. The increase in variances and the presence of additional fine-scale structures are visible in small-size ensembles (5 to 10 members) compared to the 25-member ensemble. Fine-scale structures reflect the sampling noise in the small-size ensembles, reaching a maximum in the 5-member ensembles (cf. Fig. 1, panel d). These spurious structures appear with small-size ensembles and are to be filtered later. In general, small-size ensembles correspond to fast-decreasing eigenvalues compared to the true covariance

matrix. Therefore, error variances are larger for small-size ensembles, independently of the calibration process. In addition, the variances in calibrated ensembles with more members are smaller because the inflation of the variance is a direct consequence of removing members. Hence, the calibration process better inflates the dispersion for small ensembles. Díaz-Isaac et al. (2018a) have shown the calibration process yields smaller-size ensembles to better represent model errors. Here, the variance from the 25-member ensemble remains harder to inflate by the calibration process, also less affected by sampling noise which

is likely to be less representative of the actual transport model errors.

### 3.2   Filtering of sampling noise: convergence

We show here the values of the length scales in our filter resulting from the optimality criteria, applying both Gaussian (cf. Equation 8) and non-Gaussian (cf. Equation 10) filters to the raw variances. We implemented the dichotomy algorithm proposed in Ménétrier et al. (2015b) to obtain the optimal length scale of the filter, dividing (or multiplying) the length scale by a factor of

2 until convergence. The algorithm solves for the optimal length scale of the filter by scanning the space of solutions iteratively (applying a multiplicative factor at each time step to minimize the cost function). For all our cases, the algorithm is assumed to fail at converging if the length scale is larger than 750 km. We defined this upper bound to represent about half the size of our simulation domain (square of 1,600 km wide). This large value means that the extent of noisy structures would encompass the entire domain, and therefore would not be recoverable. Here, the sampling noise is characterized by length scales of sizes

ranging from few kilometers to several hundred kilometers. In practice, the algorithm may fail at converging for two main reasons: large structures in the noise scale beyond the limit of our simulation domain, or the domain-averaged optimality





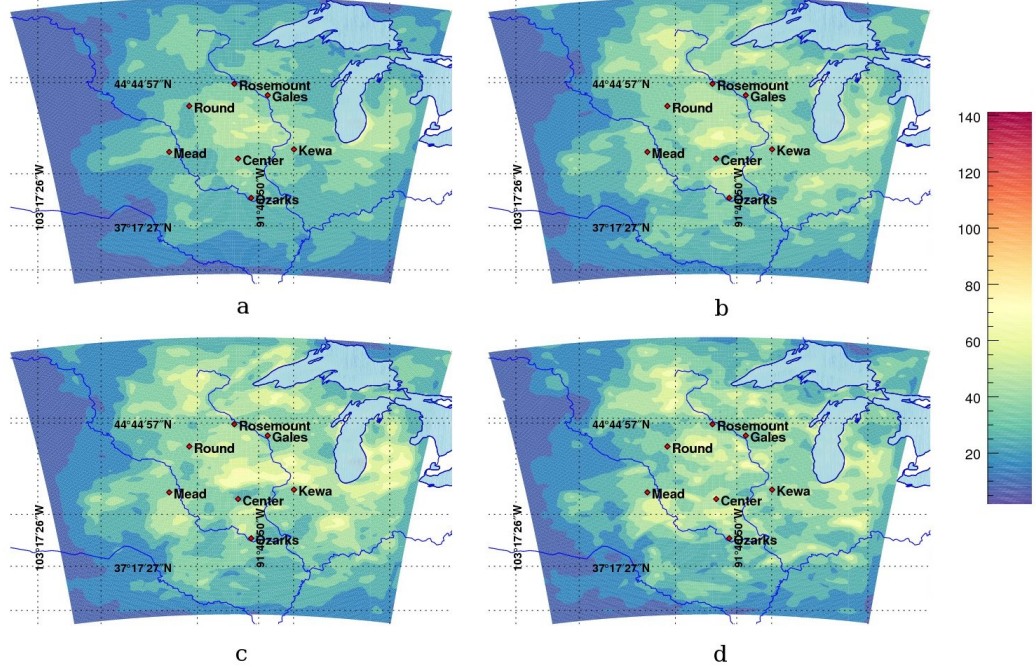

**Figure 1.** Variances of $CO_2$ in situ mole fractions (about 100m agl) (in $ppm^2$) using 25 members (upper left panel), 10 members (upper right panel), 8 members (lower left panel), and 5 members (lower right panel).

assumption is incorrect due to spatial variability in the sampling noise. In the method described by Ménétrier et al. (2015b), the ergodic assumption is necessary to diagnose robust estimators of the filter (cf. Section 2.2).

For $CO_2$ mole fractions (cf. Figure 2), the algorithm for the calibrated 25-member ensemble systematically converges to small length scales ($<50$ km), indicating that noise structures are very small in our optimal ensemble. When using the Gaussian

filter, the algorithm systematically converges for all cases except for 30% of the days with the smallest ensemble (5 members). In the non-Gaussian case, the filter converges to larger length scales, but fails at converging with the 5- and 10-member ensembles for less than 30% of the days. Typically, failures at converging are temporally coherent over periods of several days suggesting weather-related structures during which the filter repeatedly diverges. Overall, the non-Gaussian filter shows a lower rate of convergence compared to the Gaussian filter for $CO_2$ mole fractions.

For $XCO_2$ column mole fractions (cf. Figure 3), the optimal length scales are larger and the non-Gaussian filter fails frequently (about 50% of the days for 8 members or less). Even with the optimal 25-member ensembles, error structures of about 50 to 200km are filtered out, significantly larger than for the $CO_2$ mole fractions. We discuss in Section 4.2 the possible physical reasons behind the lack of convergence, possibly due to large-scale structures in the Free Troposphere or to the complexity in noise structures as XCO2 data integrate noises from different altitudes. Figure 4 shows the results of the Planetary

Boundary Layer heights (PBLH) for which both filters fail at converging for half of the days in the Gaussian case. However,



the filter shows a higher convergence rate with the non-Gaussian filter applied to 10-member ensembles. Variance noise for PBL depths present skewed distributions (not shown here) requiring the use of a non-Gaussian filter. We conclude here that 8- and 10-member ensembles are the minimum sizes with which convergence can be obtained on most days for most variables. 5-member ensembles will still be studied later on for covariances, as the localization of covariances does not depend on the filtered variances, but the low rate of convergence might limit the use of the filtered variances.

### 3.3 Variance filtering and ensemble sizes

The filtered variances shown in Figure 5, here with the Gaussian filter, for the different ensemble sizes show a better agreement both in term of spatial patterns and magnitudes among the different ensembles. The filter successfully removed noisy structures, therefore decreasing the dependence to the number of members used in each case. Despite the lack of convergence for 30% of the days with the 5-member ensemble, filtered variances at the monthly time scale show similar structures than 8- or 10-member ensembles, with nearly all of the noisy structures being removed by the filter. Compared to earlier results, the ensemble size does not seem to fundamentally limit the capacity of the filter to remove the noise, despite the lack of convergence. The variance magnitude remains slightly larger for 10 members or less, with a relative over-estimation of about 15%. Our 5-member ensemble provided the best match with only 10% higher than the 25-member filtered variances. The averaging over a whole month compensates for the lack of convergence, producing reasonable estimates of the optimal variance even with 5 members. This results suggests that climatological error variances from small-size ensembles can be a good first-approximation of the true variance when filtered correctly over most days.

### 3.4 Error variances in $CO_2$, $XCO_2$, and PBL height

We show in Figure 6 the spatial distribution of error variances from the 25-member calibrated ensemble for in situ $CO_2$ mole fractions in the PBL (100m agl), in situ $CO_2$ mole fractions in the Free Troposphere (about 5km agl), total column of $XCO_2$ dry air mole fractions, and PBL depths (in meter agl). The four variables display very distinct spatial patterns. $XCO_2$ variance spatial patterns (cf. Fig. 6, panel c) exhibit distinct maximum values located in the southwestern part of the domain, whereas high $CO_2$ variances are observed in the northeastern part of the domain for free tropospheric $CO_2$ (cf. Fig. 6, panel b) or centrally located for $CO_2$ variances in the PBL (cf. Fig. 6, panel a). Finally, PBL depth variances (cf. Fig. 6, panel d) show no indication of direct relationship between large errors in the western part of the domain and the other three $CO_2$ variables. We conclude here that no direct relationship can be utilized to construct $CO_2$ variances based on PBL depths. Similarly, maximum variances among the three $CO_2$ variables are also significantly different in distribution and magnitudes.

### 3.5 Covariance Localization: Schur and Wiener filters

Error covariances in $CO_2$ mole fractions scale with the magnitude of the surface $CO_2$ fluxes, therefore are difficult to interpret. Instead, we present here the error correlations to highlight the spatial structures inherited from the transport models, independent of the magnitude of the underlying $CO_2$ surface fluxes. We show in Figure 7 the hourly correlation structures from the





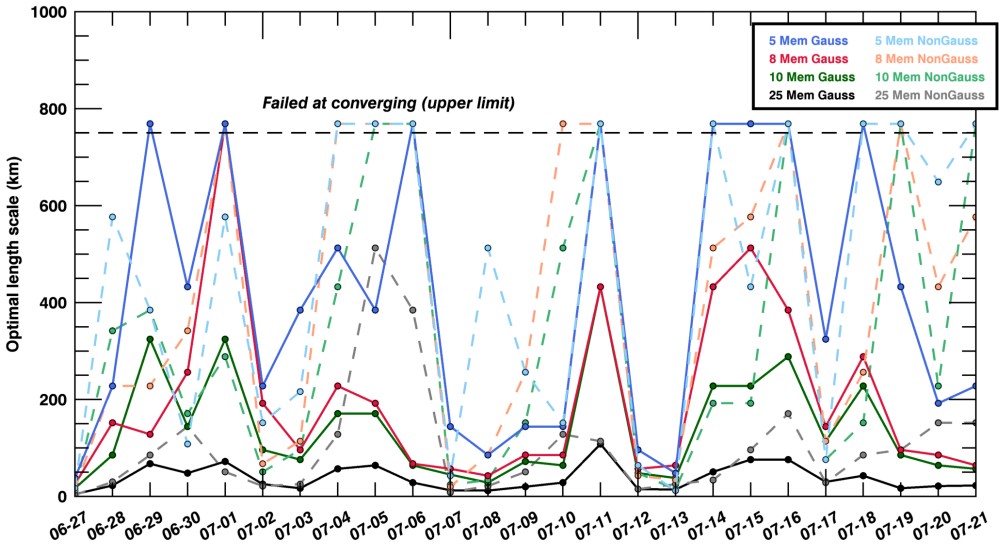

**Figure 2.** Length scale (in km) of the variance filter for in situ $CO_2$ mole fractions at about 100m agl using Gaussian (solid lines) and non-Gaussian (dash lines) equations from 27 June to 21 July, 2008.

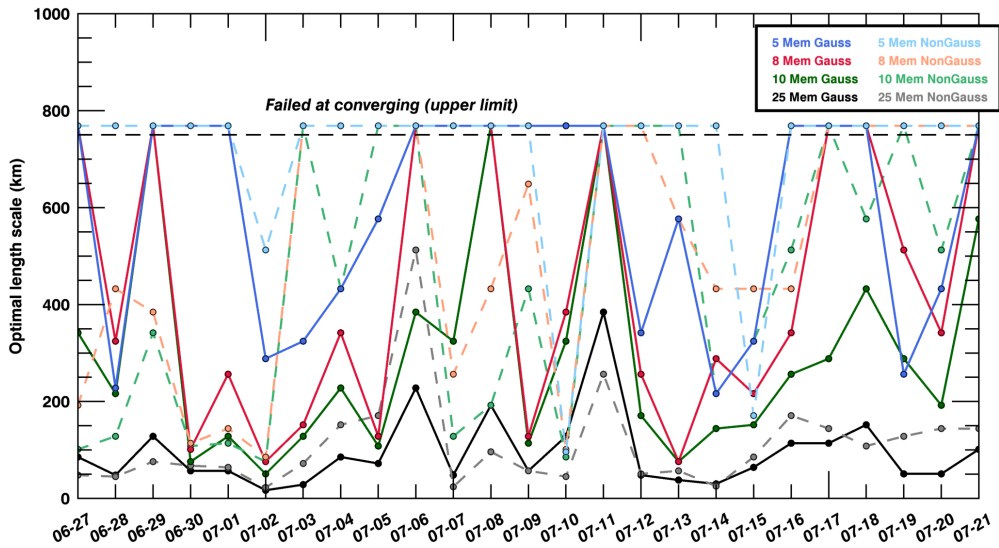

**Figure 3.** Length scale (in km) of the variance filter for $XCO_2$ total column dry air mole fractions using Gaussian (solid lines) and non-Gaussian (dash lines) equations from 27 June to 21 July, 2008.



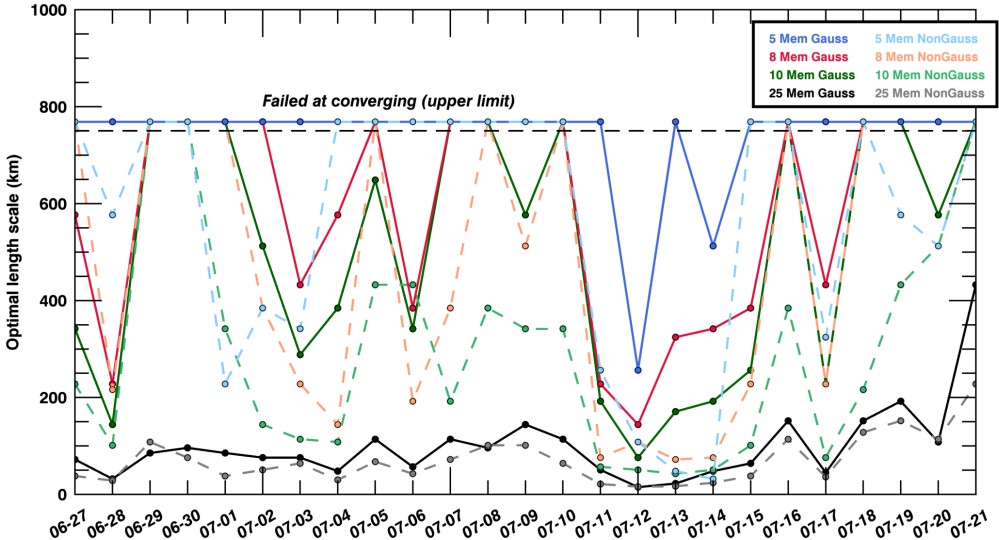

**Figure 4.** Length scale (in km) of the variance filter for Planetary Boundary Layer heights using Gaussian (solid lines) and non-Gaussian (dash lines) equations from 27 June to 21 July, 2008.

full 45-member ensemble (top row) and our 25-member calibrated ensemble (lower row) at one of the instrumented towers in the US Midwest, *i.e.* Centerville, Iowa. We applied both Schur (cf. Fig. 7, middle column) and Wiener (cf. Fig. 7, right column) filters to compare the impact of both filters on the raw correlations (cf. Fig. 7, left column). The Schur filter has less impact on the correlations compared to the Wiener filter which attenuates significantly the magnitude of the correlations for

both ensembles. In Section 2.2, the local averaging of the optimal length scale assumes that the sampling noise is spatially homogeneous (third ingredient of the methods). This assumption is required for the ergodicity assumption, therefore being a domain-averaged filtering approach. The sub-domain used for the covariance filtering (here 400x400 km$^2$) limits the spatial extent to 200 km around the observation location. The size of the domain was defined primarily for computational efficiency and based on the size of correlation structures, usually of about 100-200 km in length scale. To evaluate this assumption, we

compared the size of the sub-domain to filter the covariances with a large area of 900x900 km$^2$ for the 45-member ensemble to a smaller area of 400x400 km$^2$ for the 25-member ensemble. Filtered correlations show similar results for Schur and slightly larger values for the smaller sub-domain when applying the Wiener filter. We conclude here that the spatial local averaging has a minor impact on the results, and that our 25-member ensemble has similar spatial structures than the original 45-member ensemble, with larger correlations at short distances. We extend this analysis to the monthly time scale by showing

monthly averaged error correlations, super-imposed from different tower locations on the same map to aggregate the results at multiple locations (cf. Fig. 8). When averaged over longer time scales (cf. Fig. 8), the filtered correlations become isotropic, distributed around each location. The magnitudes remain larger with the Schur filter (cf. Fig. 8, middle panel) compared to





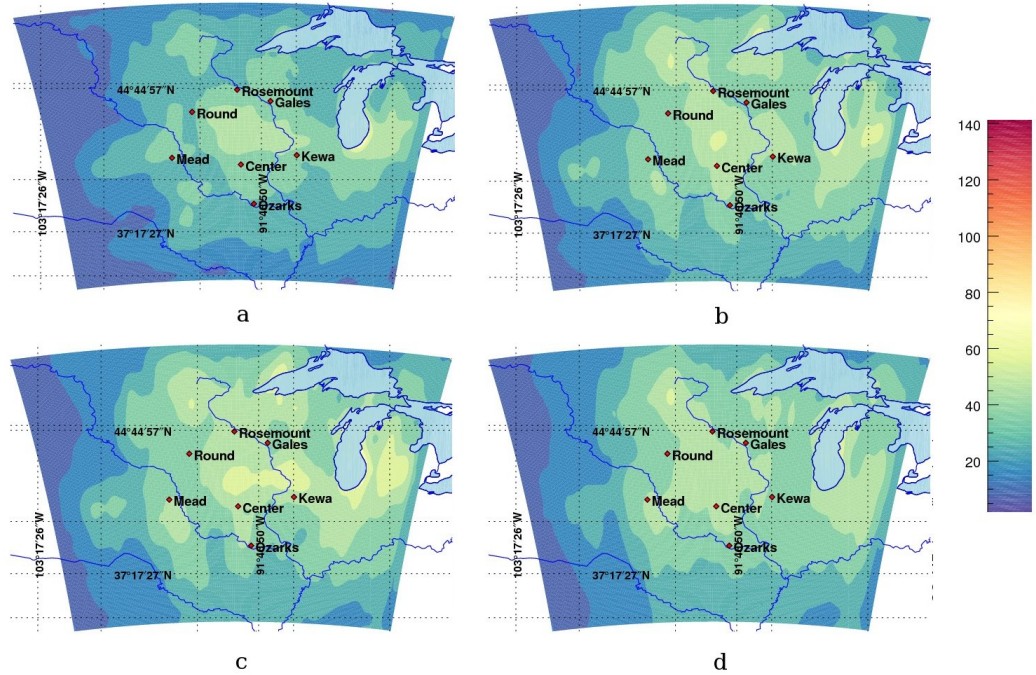

**Figure 5.** Filtered variances of in situ $CO_2$ mole fractions (about 100m agl) (in $ppm^2$) using 25 members (upper left panel), 10 members (upper right panel), 8 members (lower left panel), and 5 members (lower right panel).

the Wiener filter (cf. Fig. 8, right panel) but the differences are noticeably smaller. The unfiltered correlations (cf. Fig. 8, left panel) are noticeably larger due to noisy structures). After filtering, the spatial structures are distributed around the observation locations following a pseudo-Gaussian pattern. The magnitude of the error correlations, *i.e.* the length scale of the errors, is reduced in both cases compared to the raw correlations (cf. Fig. 8, left panel). This result confirms the sub-domain used here

5   ($400 \times 400 \, km^2$) is sufficient to represent the error correlation structures around each measurement location and describes fully the error structures.

    We show in Figure 9 the results for the different ensemble sizes using the Schur filter. 10- and 8-member ensembles show similar magnitude and patterns for the different sites, but the correlations are smaller than with the original ensemble. In comparison, the Wiener filter (cf. Fig. 10) generates consistent patterns with 25-, 10- and 8-member ensembles. In both cases,

10   the filters decreases significantly the correlations in the 5-member ensemble, revealing the difficulty of the filter to separate the noise from the actual error correlations. We discuss in Section 4.3 the relationship between meteorological and $CO_2$ variables for both error variances and error correlations.





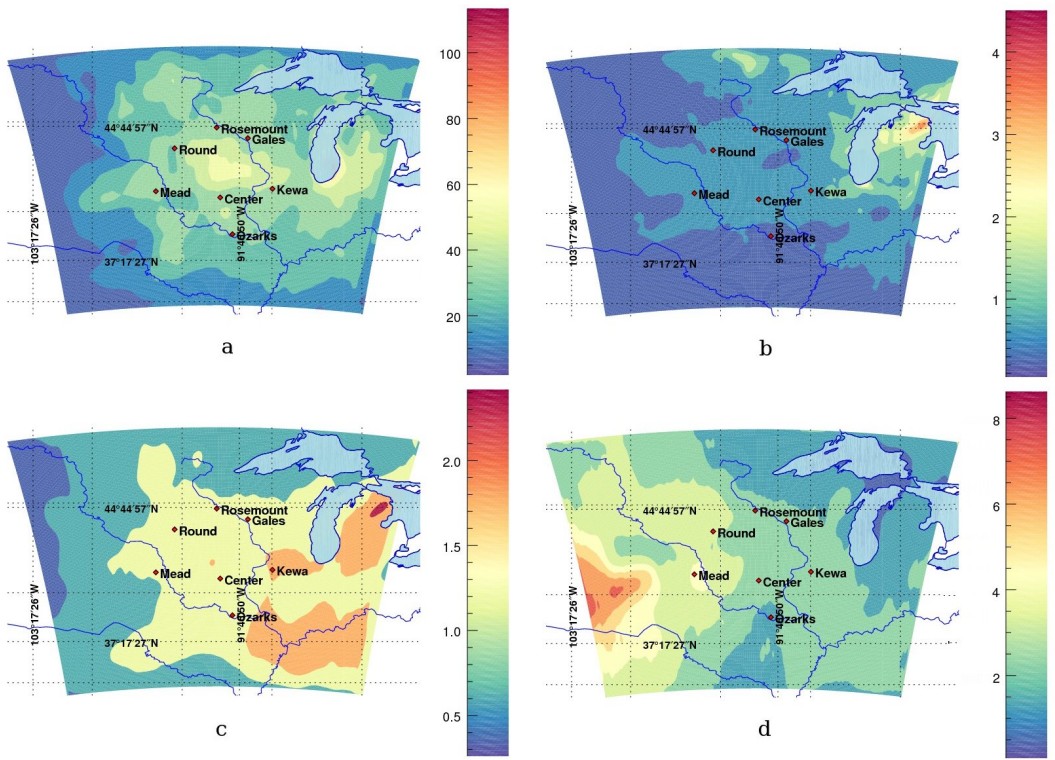

**Figure 6.** Filtered variances using the calibrated 25-member ensemble of in situ $CO_2$ mole fractions at 100m agl (in $ppm^2$) (a), in situ $CO_2$ mole fractions at 5km agl (in $ppm^2$) (b) $XCO_2$ total dry air mole fractions (in $ppm^2$) (c), and PBL height (in $m^2$) (d).

## 4  Discussion

### 4.1  Minimum size of calibrated ensembles

We discussed in Section 3.3 the dependence of the success of the variance filtering on the ensemble size. From these results, an ensemble of at least 8 to 10 members seems required to reach convergence and hence filter the error variances. However, the spatial representation of the averaged filtered variances using a calibrated 5-member ensemble (cf. Fig. 5) indicates a reasonable recovery of the error variances at the monthly time scale. Failure to converge over single days has a limited impact on the monthly mean filtered variances. We conclude here that the filter produces satisfactory results to generate first-order estimates of the $CO_2$ mole fraction errors. To achieve a systematic daily convergence, we recommend a larger number of members in the ensemble. One important point here is the calibration step performed before filtering, which optimizes the information content in each member relative to the other members. Therefore, a randomly-generated ensemble may require additional members in order to represent the actual error variances. We tested the filtering technique on random ensemble (*i.e.* uncalibrated) and found that convergence failed more frequently with 10 or less members (not shown here).

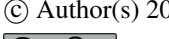


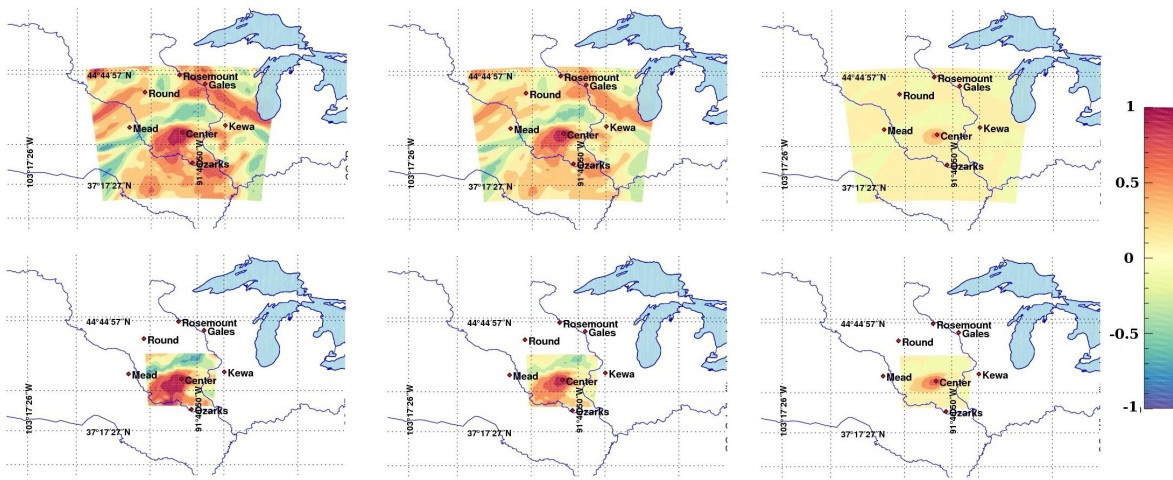

**Figure 7.** Hourly raw (left column) and filtered correlations using Schur (middle column) and Wiener (right column) filtered correlations of in situ $CO_2$ mole fractions at the Centerville tower (at about 100m agl) on June 26, 2008. The correlations are based on the original 45-member ensemble (top row) with a large subdomain (900x900 $km^2$) and the 25-member ensemble (bottom row) over the reference subdomain (400x400 $km^2$). The highlighted domain also corresponds to the domain used for the spatial averaging of the correlations in the optimality conditions.

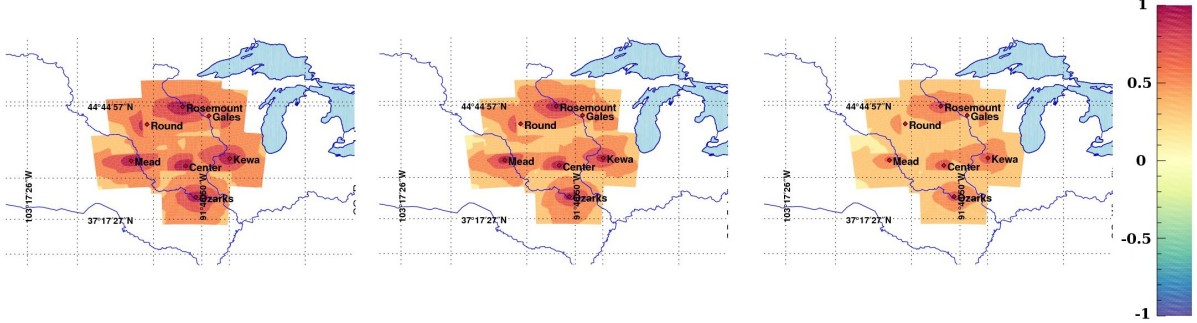

**Figure 8.** Raw (left panel) and filtered correlations of $CO_2$ in situ mole fractions at about 100m agl based on 25 members using the Schur localization (middle panel) and Wiener and Schur filter (right panel).

## 4.2 Impact of calibration on small-size ensembles

We have explored the impact of the calibration process on the error variances and covariances by filtering non-calibrated sub-ensembles of 8 and 10 members (cf. Figure 11). These random ensembles have no member in common with their calibrated counter-part, and are composed of simulations using various physics configuration randomly selected among the 45 original
5    model configurations. The optimal length scale of the variance filter is systematically lower for non-calibrated ensembles (cf.

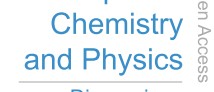
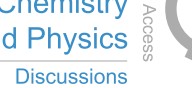

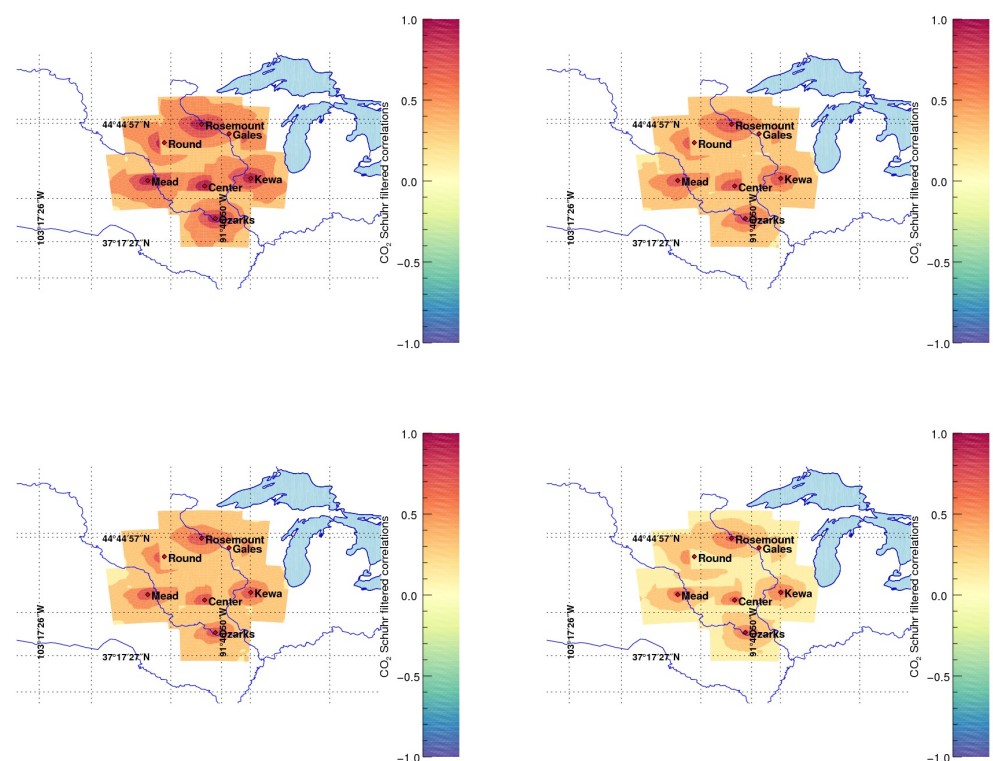

**Figure 9.** Filtered correlations (using Schur product only) of $CO_2$ in situ mole fractions (about 100m agl) (in $ppm^2$) (about 100m agl) (in $ppm^2$) using 25 members (upper left panel), 10 members (upper right panel), 8 members (lower left panel), and 5 members (lower right panel).

Fig. 11, in grey and light blue) compared to calibrated ensembles (cf. Fig. 11, in black and royal blue), suggesting lower levels of noise with larger spatial structures, similar to larger ensemble sizes (25 members or more). Because members of the calibrated ensembles were selected to maximize the information content, calibrated-ensemble members differ more from each other than non-calibrated ensemble members. As shown in Díaz-Isaac et al. (2018a), one member with higher or lower

5  PBL height statistics (usually the monthly mean model estimate) is systematically selected in order to generate calibrated ensembles with enough variance, and therefore capture the spatial and temporal variability in observed PBL heights from 14 radiosondes. Some of these members introduce different spatial structures compared to the original ensemble, increasing the spread significantly. However, the small size of our ensembles with a larger variance may affect the sampling of these structures and hence our ability to differentiate noise from actual error structures. The approach proposed by Ménétrier et al.

10  (2015a) was initially designed for purely random ensembles, which is not the case here. Some true correlation structures in the calibrated ensemble may be considered as noise by the filter. Another approach would generate an ensemble based on the



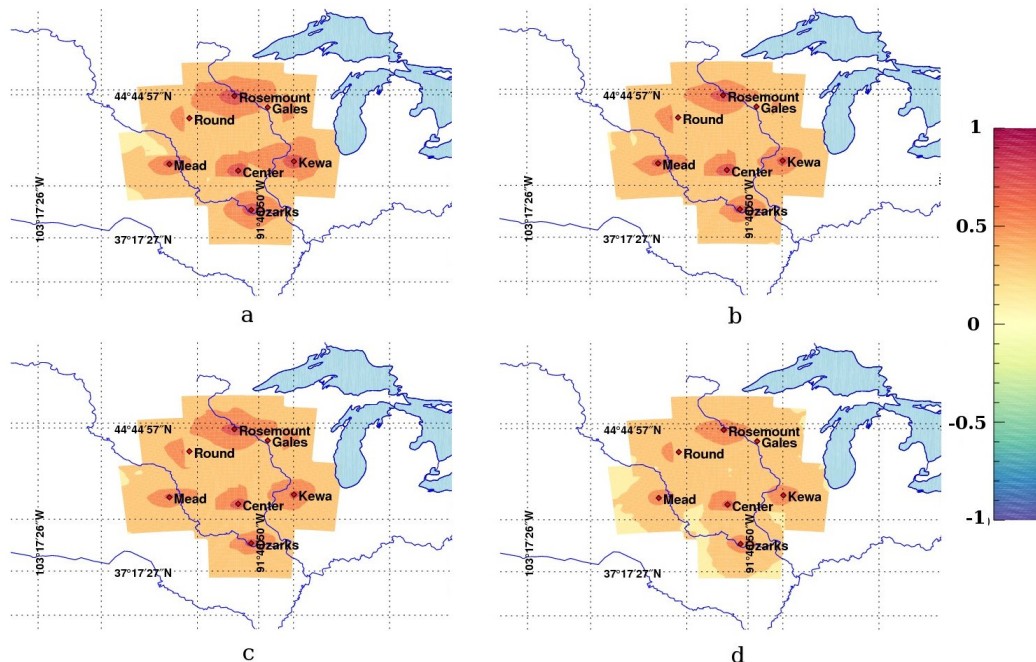

**Figure 10.** Filtered correlations (using Wiener & Schur product) of $CO_2$ in situ mole fractions (about 100m agl) (in ppm$^2$) (about 100m agl) (in ppm$^2$) using 25 members (upper left panel), 10 members (upper right panel), 8 members (lower left panel), and 5 members (lower right panel).

localized ensemble-based error covariance matrix. This approach will be developed further in future studies but is beyond the scope of this paper.

### 4.3 Spatial structures in $CO_2$ and meteorological errors

In Figure 6, significant differences in filtered error variances were observed between in situ $CO_2$ mole fractions (at different altitudes), $XCO_2$ mole fractions, and PBL height. Therefore, we conclude here that transport errors of meteorological variables are not transferable to $CO_2$ and $XCO_2$ error variances. This finding is in agreement with Miller et al. (2015) who found no direct relationship between errors in the meteorology and in situ $CO_2$ mole fractions. However, considering the covariances, the spatial structures in $CO_2$ mole fraction errors inherited from transport model errors exhibit well-defined patterns (e.g. Figure 10). By fitting a simple Gaussian function in the form of $e^{\frac{-x}{L}}$ to the filtered covariance fields, we diagnosed the characteristic length scale of the spatial error structures for the different variables, here $CO_2$ 100m-high mole fractions, the mean horizontal zonal wind component, and PBL heights. Figure 12 shows the daily length scales at the seven measurement locations over the simulation period (*i.e.* 27 June to 21 July). The length scales $L$ for the three variables increase rapidly between 27 June and 4 July from less than 100 km to 150 km or higher. As there is no long-term spin-up (re-initialization of the perturbations every



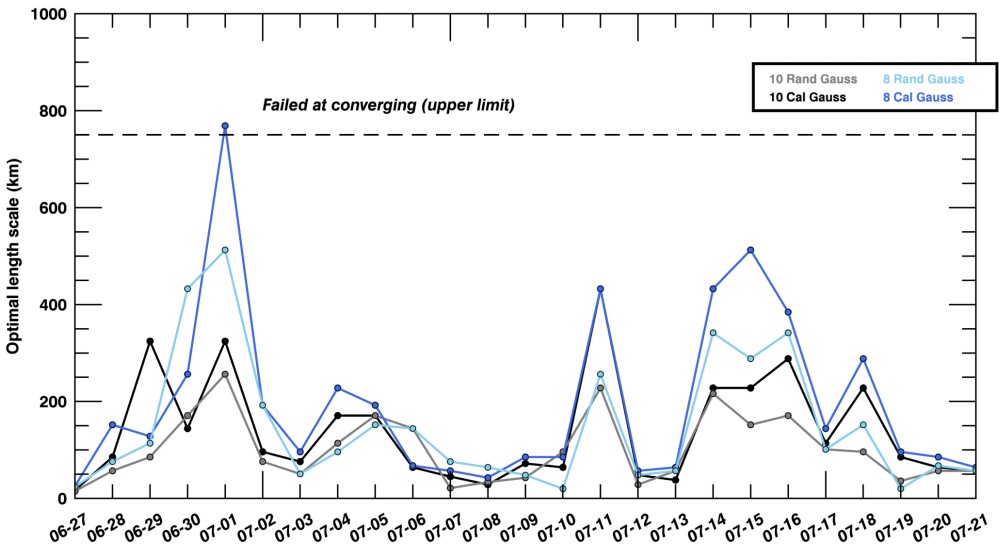

**Figure 11.** Length scale (in km) of the variance filter for 10-member random (in grey) and calibrated (in black) ensembles, and 8-member random (in light blue) and calibrated (royal blue) ensembles, for in situ $CO_2$ mole fractions at about 100m agl using Gaussian equations from 27 June to 21 July, 2008.

5 days), the asymptotic behavior is most likely due to the seasonality of the errors from early to late summer. The seasonal changes in the atmospheric dynamics impact the spatial structures in the errors for the three variables. Across the seven sites, the characteristic length scale of spatial error structures also vary significantly, in particular for PBL heights with large differences across sites. Both $CO_2$ mole fractions and the mean zonal wind component reach a maximum value over July, respectively

140 km and 120 km. The comparison of the mean length scales (lower right panel in Figure 10) highlight the differences between the three variables. Both the $CO_2$ 100m-high mole fractions and the mean horizontal zonal wind component show similar variations and converge towards the same values, but differ significantly from 29 June to July 10. We conclude here that first-order estimates for $CO_2$ spatial error correlations may be derived from meteorological error structures, in particular from the mean horizontal wind speed. But these approximations may be valid only for specific time periods. As presented in

Section 3.4, error variances in $CO_2$ and $XCO_2$ mole fractions are decoupled from PBL height errors. Here, we suggest that error correlations may be derived from wind errors but error variances should still be computed independently.

## 4.4    Modeling of error correlations

This study presents a methodology to filter the noise in error structures from a small-size ensemble. To introduce the findings of our study into an atmospheric inversion system, an additional step would be required in order to construct a regularized error

covariance matrix. For example, Lauvaux et al. (2009) proposed to model the error structures using a diffusion equation able to





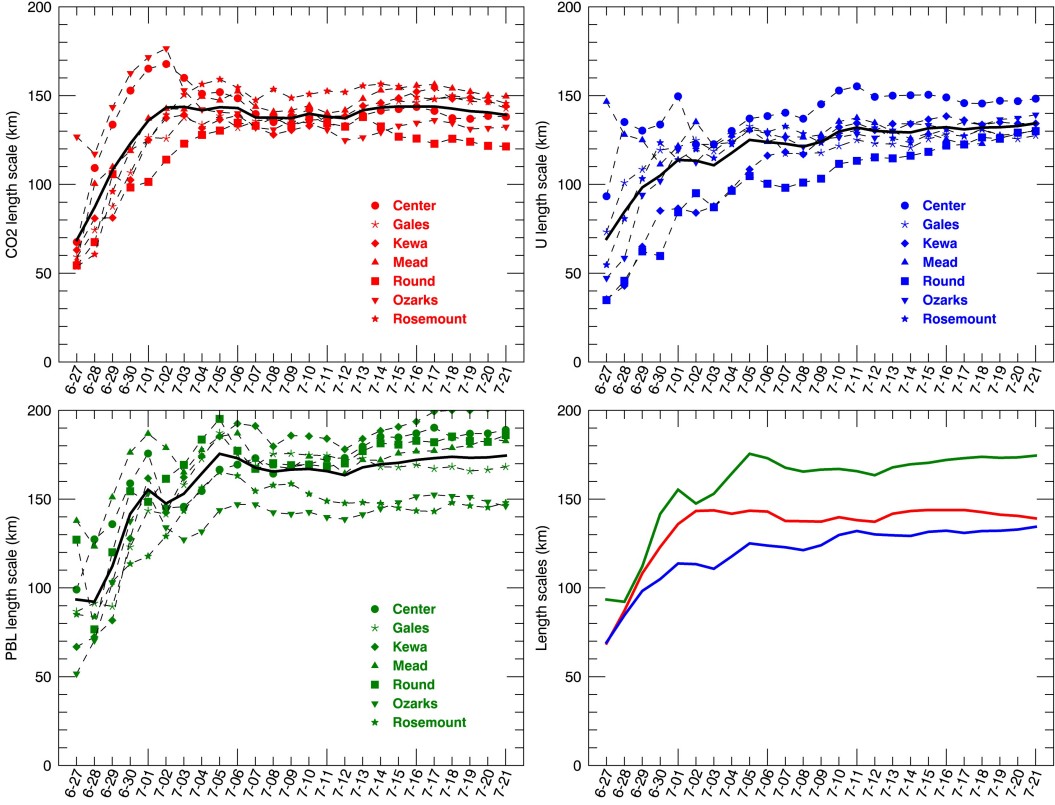

**Figure 12.** Characteristic length scales $L$ of filtered error correlations (from the 25-member ensemble) using an exponential function ($e^{\frac{-x}{L}}$) for $CO_2$ 100m-high mole fractions (upper left panel), the mean horizontal zonal wind component ($U$; upper right panel), PBL heights (lower left panel), and the mean values for the three variables (lower right panel) for every day of the simulation period (27 June to 21 July).

represent anisotropic structures, following the methodology described in Pannekoucke et al. (2008). Here, we presented a local filter able to remove the noise in the error structures. The diagnosed error structures can be approximated by different spatial functions of varying degree of complexity, further regularized to generate positive-definite error covariance matrices. This next step is beyond the scope of this paper but will be conducted in the future to generate an efficient model of the corresponding

5    error covariances based on our current results.

## 5    Conclusions

We have diagnosed the error variances and the spatial error structures from our mesoscale transport models at daily and monthly time scales. Applied to both $CO_2$ mole fractions and meteorological variables, we implemented a cost-effective filtering technique currently used in meteorological data assimilation systems (Ménétrier et al., 2015b) to describe spatial error structures

10    using a small-size ensemble. The approach remains affordable for multi-year inversions of sources and sinks at continental




or regional scales. The removal of noisy structures in our small-size ensembles is evaluated by comparison to larger-size ensembles, both the original 45-member ensemble and our optimal calibrated sub-ensemble of 25 members. A second filtering approach for error covariances was successfully applied using the Wiener filter, producing similar results compared to the Schur filter over the 1-month simulation period. Differences were noticeable at shorter time scales (*i.e.* daily). The spatial distribution

of error variances and spatial error structures are recoverable from small-size ensembles of 8 to 10 members, providing a more realistic representation of transport errors in future mesoscale inversions of $CO_2$ fluxes. We noted that error variances of in situ $CO_2$ mole fractions and total column $XCO_2$ differ significantly, even when varying the altitudes or considering PBL height error structures. We conclude that error variances for remote sensing observations need to be quantified independently of PBL or free tropospheric mole fractions.

We have discussed the potential use of meteorological error structures such as the mean horizontal wind to approximate spatial error correlations of in situ $CO_2$ mole fractions. The seasonal variations in wind, PBL height, and in situ $CO_2$ mole fractions are highly correlated, while the typical length scales in error structures vary from 100km to 150km in the middle of summer depending on the variable. We conclude here that meteorological error structures may provide a first-order estimation of correlation length scales in $CO_2$ inversions when no ensemble of $CO_2$ simulations is available.

**6   Code availability**

The code is accessible under request by contacting the corresponding author (tul5@psu.edu).

**7   Data availability**

The model simulation outputs are available under request by contacting the corresponding author (tul5@psu.edu).

**Appendix A:   Calibration of the reference 25-member ensemble**

In this study, we generate a reference ensemble to evaluate the sampling noise in the ensembles of smaller sizes. The original 45-member ensemble, uncalibrated, cannot be used as a reference as it under-estimates the model errors. The reference ensemble needs to include enough members to limit the sampling noise. Based on the same original ensemble of 45-members as in Díaz-Isaac et al. (2018a), we reduce it to a calibrated 25-member ensemble using the Simulated Annealing (SA) algorithm. The selection of the optimal calibrated ensemble is based on three meteorological variables (i.e., wind speed, wind direction

and PBL height) and follows the same procedure described in Díaz-Isaac et al. (2018a). The SA for 25 members uses 40,000 iterations to reach convergence, significantly larger than the 20,000 iterations for 10-, 8- and 5-member ensemble. The same criteria used by Díaz-Isaac et al. (2018a) was applied to the selection process of the calibrated 25-member ensemble, improving the flatness of the rank histograms (Fig. A1). This selection is based on two criteria. First, we selected all the 25-member sub-ensembles with a rank histogram score smaller than six for each individual meteorological variable. In a second step, we filtered

out the 25-member ensembles accepted by the SA algorithm but corresponding to a bias (i.e. mean model-data mismatch over





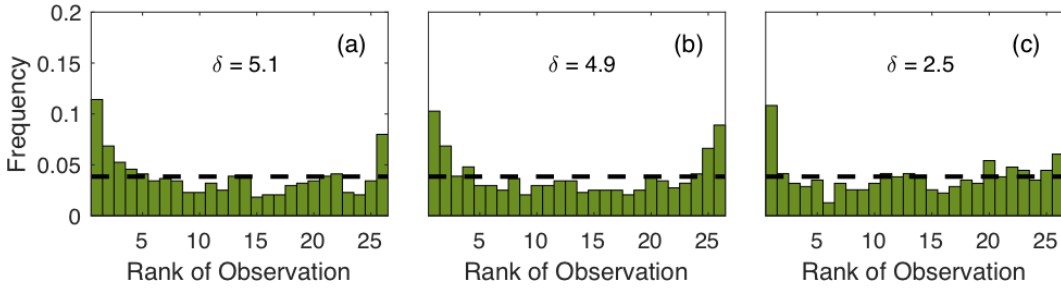

**Figure A1.** Rank histograms of the 25-member ensemble after calibration for wind speed (a), wind direction (b) and Planetary Boundary Layer Height (c) over 35 days (June 18 to July 21, 2008) using 14 rawinsonde sites available over the US Midwest. The horizontal dashed line corresponds to the ideal value for a flat rank histogram with respect to the number of members. The rank histogram score ($\delta$) defines the flatness of the rank histogram, 1 being the ideal value.

25 days) larger than the bias in the original 45-member ensemble. These criteria are applied to the three meteorological variables. This procedure is described in more details in Díaz-Isaac et al. (2018a). The rank histograms in Figure A1 show a limited under-dispersion of the 25-member ensemble, significantly reduced after calibration from 6.1, 6.2, 3.2 to 5.1, 4.9, and 2.5 for wind speed, wind direction, and PBLH respectively.

5   *Author contributions.* The WRF-Chem simulations were performed by L.I. Díaz-Isaac and T. Lauvaux; the filtering technique was coded by M. Bocquet and T. Lauvaux based on the work of B. Ménétrier; the concept and ideas were designed by M. Bocquet, T. Lauvaux, and N. Bousserez; the manuscript was prepared by T. Lauvaux, L.I. Díaz-Isaac, N. Bousserez and M. Bocquet.

*Competing interests.* The authors declare that they have no conflict of interest

*Acknowledgements.* This research was supported by National Aeronautics and Space Administration (NASA) Terrestrial Ecosystem and
10  Carbon Cycle Program (grant #NNX11AE79G), by the NASA's Earth Venture Program Atmospheric Carbon and Transport (ACT)- America (grant #NNX15AG76G), the Alfred P. Sloan Graduate Fellowship, by the NASA Carbon Monitoring System program (grant #NNX13AP34G), and by the National Oceanic and Atmospheric Administration (grant #NA14OAR4310136). CEREA is a member of *Institut Pierre-Simon Laplace* (IPSL).



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
