# Peer review of "Diagnosing spatial error structures in CO2 mole fractions and XCO2 column mole fractions from atmospheric transport"

_Atmospheric Chemistry and Physics, 2018_

## Referee Comment (RC1) · Benjamin Ménétrier (Referee) · 25 Feb 2019

**1  General comments**

In this article, the authors use covariance filtering methods recently developed in the NWP data assimilation context and adapt them in the atmospheric transport framework. Their goal is to get a better estimation of the forecast error spatial structures for in situ carbon dioxide mole fractions and total column dry air mole fractions. I think that it is a very good idea to import methods from one domain to another, and I congratulate the authors for this effort.

[Figure]

I am not a specialist of atmospheric chemistry, so I will not be able to assess the relevance of the experimental setup used in this paper. In my comments, I will focus on the covariance filtering aspects, since the authors uses the theory that I have developed for my PhD thesis.

I think that some parts of the paper need major clarifications regarding the use and implementation of filtering methods. Detailed remarks are listed hereafter.

**2   Specific comments**

1. I think that there is a mistake in the definition of the criterion $\mathcal{L}(\mathbf{F})$ line 14. The criterion should be a scalar, so the transpose sign should be put after the first parentheses: $\mathcal{L}(\mathbf{F}) = \mathbb{E}[(\mathbf{x}^\star - \mathbf{F}\widetilde{\mathbf{x}})^\mathsf{T}(\mathbf{x}^\star - \mathbf{F}\widetilde{\mathbf{x}})]$. As a consequence, the partial derivative of $\mathcal{L}$ with respect to $F_{ij}$ is: $\frac{\partial \mathcal{L}}{\partial F_{ij}} = -2\mathbb{E}[(x_i^\star - \widehat{x}_i)\widetilde{x}_j]$. Setting this partial derivative to zero yields: $\mathbb{E}[(x_i^\star - \widehat{x}_i)\widetilde{x}_j] = 0$ for all $i$ and $j$, so: $\mathbb{E}[(\mathbf{x}^\star - \widehat{\mathbf{x}})\widetilde{\mathbf{x}}^\mathsf{T}] = 0$, which is the result given at line 17. I am not sure whether the total variation of the criterion $\delta\mathcal{L}$ is really relevant here.

2. I don't really understand why the authors assume that the dichotomy algorithm fails at converging if the filtering length-scale becomes larger than 750 km. Indeed, as shown in Ménétrier *et al.* (2015a), appendix C, the filtering length-scale verifying the optimality criterion for a homogeneous and isotropic filter always exists and is unique. As explained in section 8 of this paper, the value of this optimal filtering length-scale can be related to two ratios:

   - the signal-to-noise ratio for the variance (related to the true variance values and the sample size),
   - the ratio between the variance signal and noise spatial variations length-scales (the noise length-scale being related to the forecast error correlation

length-scale, see Raynaud *et al.* (2009))

If the optimal filtering becomes very large, leading to a constant value for filtered variances, it does not mean that the dichotomy algorithm is failing. It is a sign that the noise amplitude is too large compared to the signal variations and/or that the variance signal and noise length-scales are mixed. As a consequence, the filter prefers to filter out all spatial variations and keeps the mean value only (which might be the best option). I suggest a rewriting of the discussion about the convergence failures.

3. The covariance filtering theory leads to fractions of polynomials with factors $(N - 3)$ at the denominator, so 4 members is the absolute minimum ensemble size for this theory. I think that a 5-member ensemble is still too small to get significant results, and I would suggest to use ensembles of size 8-10 at least.

4. For the localization of correlation functions, the Schur filters given by equations (9) and (11) are not positive definite functions, as mentioned in Ménétrier *et al.* (2015b). Thus, a fit of the raw localization function with a positive definite function is required before applying the localization. It seems that the raw localization function was used in this paper, which is problematic. Also, it is not clear how the spatial and angular average was performed to estimate the statistical expectations of equations (9) and (11) via an ergodicity assumption.

5. For Figures 8, 9 and 10, the small domains overlap, which makes difficult to see the functions shapes. It would be better to have separate boxes as in Figure 13 of Ménétrier *et al.* (2015b). Moreover, it would be more helpful to plot the localization functions, rather than the localized correlation functions.

---

## Referee Comment (RC2) · Anonymous Referee #2 · 10 May 2019

This manuscript describes a method of spatially filtering small ensembles of model forecasts. Ensembles of greenhouse gas (GHG) model simulations are useful for understanding transport errors whose covariances are needed when estimating surface fluxes using inverse modelling. Since GHG flux estimates are frequently needed on long (multi-year or decadal) time scales, large ensembles may be prohibitively expensive. Thus if a small ensemble is reliable, it could be useful for approximating transport error covariances if sampling errors arising from the small ensemble size can be filtered. Thus, the topic of this manuscript has important applications to GHG flux inversion systems. The application of the variance and correlation filtering methods from meteorology to GHG forecast ensembles is novel and, as shown in the manuscript, can

reveal relationships (or lack thereof) between meteorological variables and CO2. However, conclusions regarding minimum ensemble sizes for estimating spatial variances or correlations should be carefully qualified. With small ensemble sizes, there is a lack of convergence from day to day that can be ameliorated with temporal averaging (or temporal filtering). This is akin to increasing the ensemble size with members from different days. Therefore, the authors are requested to review the manuscript and ensure all qualifications are presented when recommending ensemble sizes. Because I am not familiar with the iterative filtering methods in this work, my comments are mainly confined to the application of the filtering methods to carbon cycle science.

Specific comments

1. While the idea of filtering small ensembles is very appealing, I am not comfortable with the criterion for success being solely based on the convergence of the schemes. The quality of the ensemble can be checked by issuing forecasts and comparing these to observations. If this is difficult to do, can the methods be checked using simulated observations?

2. P1, L12-14: "We conclude that. . ." This statement needs qualification. On P14, L7-10 the authors note that for daily convergence a larger number of members is needed. The smaller ensemble seemed to work, only if additional temporal averaging was done.

3. P6, L12: It is stated that a Schur filter will be used for correlations but in line 17, it is applied to covariances not correlations.

4. P6, eq. 8: It would be useful to define (in words) the symbol on the left side of (8). For example, is it an optimality criterion? Ménétrier et al. (2015a) also does not define this symbol so it would be useful to add a few words here.

5. P7, eq.12: What is $\bar{E}$, in the denominator? The overbar is never defined. Since it does not appear in (13), it may be a typo.
6. P7, L17-18: Using the stated assumption that the true covariance is approximated by the sample covariance (i.e. $B_{ij}^* = \tilde{B}_{ij}$) will not yield (13) from (12). Instead, it will yield 1. If the true covariance $B_{ij}^* = E(\tilde{B}_{ij})$, then (13) can result.

7. P8, L14-15: I don't follow this argument. The trace of a covariance matrix (or sum of variances) equals the sum of its eigenvalues. Fast decreasing eigenvalues suggests you are talking about the spectrum of eigenvalues. The spectrum of eigenvalues of the correlation matrix does imply spatial filtering so a steep spectrum implies smoother fields. But what does the spectrum of eigenvalues of covariance matrix say about variance, as a function of ensemble size? Also, why should small ensemble sized in general have larger variances than larger ensemble sizes? Some references or mathematical derivation may be helpful here.

8. P8, L17: "dispersion" of what?

9. P9, L7-8: "Typically, ..." In Fig. 2, the 25-member ensemble has much greater temporal consistency of the optimal length scale. Since temporal correlations also suffer from sampling error, particularly with small ensemble sizes, how much can one infer about the temporal variability of the length scale with ensemble sizes of 5 or so?

10. P10, L7: "Better". Better than what? Presumably the statement refers to a comparison with Fig. 1.

11. Fig. 5 caption: What is the time? Presumably these are monthly averages.

12. P10, L16-17: This suggests you should be able to increase ensemble size using different days.

13. P10, L21-22: "XCO2 variance spatial patterns (Fig. 6c) exhibit distinct maximum values located in the southwestern part of the domain," I don't see this. I see

maxima (orange and red regions) in the eastern part of the domain. Also, the red region south of Lake Huron is similar to that seen in CO2 at 5 km in Fig. 6b. This is somewhat reassuring.

14. P14, L9-10: "One important point here is the calibration step performed before filtering..." The calibration steps in Diaz-Isaac et al. (2018a) and Appendix A were done over 18 June -21 July 2008 which encompasses the period of optimal filtering (Figs. 2-4). How applicable would this calibrated ensemble be to a different time period? My guess is that the calibration might yield a different selection of members for a different time period as the meteorology changes. So would you need to redo the calibration continually with time? Some addition discussion of the calibration process as a function of time might be useful here or in section 4.2.

15. P18, L5: "Figure 10" should be "Figure 12"

16. Figure 12: Please add a legend to the bottom right panel of this figure, to label the curves. One can see which curve is which variable by comparing to the other panels, but a legend would be much more convenient for the reader.

17. P20, L4-6: As discussed in comment 1, the same qualifications need to be made here. Specifically, is this conclusion valid only in the context of additional temporal averaging?

Technical comments

1. P1, L4: "of which" should be "whose"

2. There are many instances of "fail at converging" which should be replaced by "fail to converge". i.e. P8L27, P8L30, P9L6, P9L15.

3. P10, L9: "dependence to" should be "dependence on"

4. P12, L13: "similar. . .than" should be "similar. . .to"

5. P13, L10: "difficulty" or "inability"?

---

## Author Comment (AC1) · 29 Aug 2019

General comments In this article, the authors use covariance filtering methods recently developed in the NWP data assimilation context and adapt them in the atmospheric transport framework. Their goal is to get a better estimation of the forecast error spatial structures for in situ carbon dioxide mole fractions and total column dry air mole fractions. I think that it is a very good idea to import methods from one domain to another, and I congratulate the authors for this effort.

I am not a specialist of atmospheric chemistry, so I will not be able to assess the relevance of the experimental setup used in this paper. In my comments, I will focus on the

covariance filtering aspects, since the authors use the theory that I have developed for my PhD thesis. I think that some parts of the paper need major clarifications regarding the use and implementation of filtering methods. Detailed remarks are listed hereafter.

Authors: We thank the reviewer for constructive and helpful comments that improved the quality of this study.

Specific comments 1. I think that there is a mistake in the definition of the criterion L(F) line 14. The criterion should be a scalar, so the transpose sign should be put after the first parentheses

Authors: The equations have been corrected accordingly. Thank you for spotting this.

2. I don't really understand why the authors assume that the dichotomy algorithm fails at converging if the filtering length-scale becomes larger than 750 km. Indeed, as shown in Ménétrier et al. (2015a), appendix C, the filtering length-scale verifying the optimality criterion for a homogeneous and isotropic filter always exists and is unique. As explained in section 8 of this paper, the value of this optimal filtering length-scale can be related to two ratios: â˘Ać the signal-to-noise ratio for the variance (related to the true variance values and the sample size), â˘Ać the ratio between the variance signal and noise spatial variations length-scales (the noise length-scale being related to the forecast error correlation length-scale, see Raynaud et al. (2009))

If the optimal filtering becomes very large, leading to a constant value for filtered variances, it does not mean that the dichotomy algorithm is failing. It is a sign that the noise amplitude is too large compared to the signal variations and/or that the variance signal and noise length-scales are mixed. As a consequence, the filter prefers to filter out all spatial variations and keeps the mean value only (which might be the best option). I suggest a rewriting of the discussion about the convergence failures.

Authors: We have replaced the term "failure" by "convergence beyond our 750-km threshold" and clarified the description of the convergence. We set the threshold to

750km assuming that length scales beyond that value correspond to filtering structures extending beyond our simulation domain of 1,600km by 1,600km, hence not filtering spurious noise as initially intended. The algorithm may converge to large values but such dimensions have no physical meaning. As the reviewer pointed out, these large values correspond to low signal-to-noise ratios, or to similarities between noise structures and true variance structures. Our initial terminology was incorrect because the filter does converge. The text has been modified through the entire paper and the terminology changed.

3. The covariance filtering theory leads to fractions of polynomials with factors (N − 3) at the denominator, so 4 members is the absolute minimum ensemble size for this theory. I think that a 5-member ensemble is still too small to get significant results, and I would suggest to use ensembles of size 8-10 at least.

Authors: We now recommend 8 to 10 members in a more general case. The introduction and discussion sections have been updated accordingly.

4. For the localization of correlation functions, the Schur filters given by equations (9) and (11) are not positive definite functions, as mentioned in Ménétrier et al. (2015b). Thus, a fit of the raw localization function with a positive definite function is required before applying the localization. It seems that the raw localization function was used in this paper, which is problematic. Also, it is not clear how the spatial and angular average was performed to estimate the statistical expectations of equations (9) and (11) via an ergodicity assumption.

Authors: We agree with the reviewer that this step is currently missing. Considering that this study only looks at the filter to diagnose variances and correlations, we decided to skip the regularization for now, assuming that this part is critical when constructing the error covariance matrix out of the filtered variances and covariances. In a future study, we will regularize the solution before using our diagnosed error structures in an actual $CO_2$ flux inversion. We added some text in the last section to disclose that problem for

future studies.

5. For Figures 8, 9 and 10, the small domains overlap, which makes difficult to see the functions shapes. It would be better to have separate boxes as in Figure 13 of Ménétrier et al. (2015b). Moreover, it would be more helpful to plot the localization functions, rather than the localized correlation functions.

Authors: Unfortunately, the overlap between towers is inevitable due to the distances between sites. To generate a figure similar to figure 13 of Ménétrier et al. (2015b), the map would need to be distorted with 7 separate blocks corresponding to the 7 measurement locations used here. Instead, to present more clearly the results of our study, we added some text with a reference to the figure 12 in which length scales are presented more clearly for each tower and for each day. "We present the localized correlation length scales for each tower and for each day in Fig. 12 (upper left panel). For both Center and Mead, length scales are noticeably larger than for the other towers and decrease rapidly until July 2nd, before converging back to the same values diagnosed for other measurement sites. The differences across towers suggest local differences in error correlations, even across the same region for a single day (up to 70 km across our sites). These differences correspond to the beginning of summer, when both weather and ecosystem fluxes vary rapidly especially in agricultural areas."

Authors: Related to localization functions, we did not include localization functions voluntarily to avoid confusion between error correlations and localization functions. This study aims at presenting the filtered error correlations for future applications, with less emphasis on the filtering method itself.

---

## Author Comment (AC2) · 29 Aug 2019

This manuscript describes a method of spatially filtering small ensembles of model forecasts. Ensembles of greenhouse gas (GHG) model simulations are useful for understanding transport errors whose covariances are needed when estimating surface fluxes using inverse modelling. Since GHG flux estimates are frequently needed on long (multi-year or decadal) time scales, large ensembles may be prohibitively expensive. Thus if a small ensemble is reliable, it could be useful for approximating transport error covariances if sampling errors arising from the small ensemble size can be filtered. Thus, the topic of this manuscript has important applications to GHG flux inversion systems. The application of the variance and correlation filtering methods from meteorology to GHG forecast ensembles is novel and, as shown in the manuscript, can reveal relationships (or lack thereof) between meteorological variables and CO2. However, conclusions regarding minimum ensemble sizes for estimating spatial variances or correlations should be carefully qualified. With small ensemble sizes, there is a lack of convergence from day to day that can be ameliorated with temporal averaging (or temporal filtering). This is akin to increasing the ensemble size with members from different days. Therefore, the authors are requested to review the manuscript and ensure all qualifications are presented when recommending ensemble sizes. Because I am not familiar with the iterative filtering methods in this work, my comments are mainly confined to the application of the filtering methods to carbon cycle science.

Authors: We thank the reviewer for constructive and helpful comments on the application of the filtering approach. We clarified our conclusions to consider all aspects of small ensemble sizes and to improve the quality and readability of this study.

Specific comments 1. While the idea of filtering small ensembles is very appealing, I am not comfortable with the criterion for success being solely based on the convergence of the schemes. The quality of the ensemble can be checked by issuing forecasts and comparing these to observations. If this is difficult to do, can the methods be checked using simulated observations?

Authors: We fully agree with the reviewer that the ensemble should be evaluated with atmospheric observations. An important part of this study relies on the work of Diaz-Isaac et al. (2018) who calibrated the ensemble using meteorological measurements from radiosondes. Assuming we have generated a fairly reliable ensemble, we diagnosed error correlations that would also require further evaluation with spatially-dense datasets from, e.g., aircraft campaigns. Gerbig et al. (2003) used atmospheric measurements from the COBRA campaign over North America to diagnose the error correlations. Here, we have no such campaign as we intend to diagnose daily error correlations over a large domain. Existing campaigns such as ACT-America (2016-2019) will

be of great use to evaluate the filtered structures. We have added a discussion on this topic in the paper. The last section is now entitled "Evaluation and modeling of error correlations":

"The evaluation of the filtered structures would benefit from dense measurement campaigns sampling spatial structures across large domains, such as the Atmospheric Carbon and Transport (ACT)-America campaigns. Previous studies have shown the utility of aircraft measurements to diagnose error correlations (Gerbig et al., 2003) but the separation of spatial structures induced by surface flux errors and atmospheric transport errors remain challenging in order to construct observation error covariance matrices. The combination of ensemble systems such as Ensemble Kalman Filter (EnKF) systems and intensive aircraft campaigns will provide additional insights to evaluate filtering approaches (e.g. Chen et al., 2019)."

2. P1, L12-14: "We conclude that. . .". This statement needs qualification.

Authors: The sentence was modified to explain our findings more precisely.

On P14, L7-10 the authors note that for daily convergence a larger number of members is needed. The smaller ensemble seemed to work, only if additional temporal averaging was done.

Authors: We have clarified the fact that 5-member ensembles seem too small to recover daily structures, and also commented that 5 members is very close to the theoretical limit of 4 members.

"However, the spatial representation of the averaged filtered variances using a calibrated 5-member ensemble (cf. Fig. 5) indicates a reasonable recovery of the error variances at the monthly time scale but not at the daily time scale. Theoretically, the minimum number of members for the covariance filtering is four, based on the Eq. 11 with a factor (N-3) in the denominator, which shows that 5-member ensembles are close to this limit and are not recommended in a more general context. The application

here suggests 5-member ensembles are acceptable but 8- to 10-member ensembles would be a minimum both in practice and in theory over different seasons and regions."

3. P6, L12: It is stated that a Schur filter will be used for correlations but in line 17, it is applied to covariances not correlations.

Authors: Because we haven't regularized the covariances, we applied the filter to covariances and generated correlations at the end. We clarified that point in the last section of the paper.

"In this study, we acknowledge here that we have applied Schur and Wiener filters using the raw filtering matrices (Eqs. 9, 11, and 6), that may not be semi-positive definite, requirement for getting semi-positive definite regularized covariances.Future studies should include an additional step by adding a regularization of the covariances before filtering."

4. P6, eq. 8: It would be useful to define (in words) the symbol on the left side of (8). For example, is it an optimality criterion? Ménétrier et al. (2015a) also does not define this symbol so it would be useful to add a few words here.

Authors: We have clarified the meaning of CiG the optimality criterion in the Gaussian case.

5. P7, eq.12: What is ÄŠ, in the denominator? The overbar is never defined. Since it does not appear in (13), it may be a typo.

Authors: We corrected this mistake. Thank you for spotting this.

6. P7, L17-18: Using the stated assumption that the true covariance is approximated by the sample covariance (i.e. $B_{ij}$âĹŮ = $B\grave{I}\check{C}_{ij}$âĹŮ) will not yield (13) from (12). Instead, it will yield 1. If the true covariance $B_{ij}$ = E( $B\grave{I}\check{C}_{ij}$ ), then (13) can result.

Author: Actually, there was a typo in between (12) and (13) : it should be $B_{ij}$âĹŮ = E[$B\grave{I}\check{C}_{ij}$]. As a consequence, Eq.(13) is correct and is actually exact (not an approximation at this stage – but will later be assessed using the local averages). Thank you for seeing the inconsistency. This has been corrected.

7. P8, L14-15: I don't follow this argument. The trace of a covariance matrix (or sum of variances) equals the sum of its eigenvalues. Fast decreasing eigenvalues suggests you are talking about the spectrum of eigenvalues. The spectrum of eigenvalues of the correlation matrix does imply spatial filtering so a steep spectrum implies smoother fields. But what does the spectrum of eigenvalues of covariance matrix say about variance, as a function of ensemble size? Also, why should small ensemble sized in general have larger variances than larger ensemble sizes? Some references or mathematical derivation may be helpful here.

Authors: We decided to remove the statement about the spectrum for a lack of a clear explanation and demonstration. We also clarified the comment about raw error variances: "The range of values for error variances increases for small-size ensembles, independently of the calibration process"

8. P8, L17: "dispersion" of what?

Authors: We clarified the text: "...dispersion of members from the mean..."

9. P9, L7-8: "Typically, . . ." In Fig. 2, the 25-member ensemble has much greater temporal consistency of the optimal length scale. Since temporal correlations also suffer from sampling error, particularly with small ensemble sizes, how much can one infer about the temporal variability of the length scale with ensemble sizes of 5 or so?

Authors: This is possible. We suspect that during these periods, the spatial scale of the sampling is similar to true variances, or that the noise is larger than the true variances with small-size ensembles. The text was clarified accordingly.

10. P10, L7: "Better". Better than what? Presumably the statement refers to a comparison with Fig. 1.

Authors: We meant that the filtered variances are in better agreement with the 25-

member ensemble used as a reference. We clarified the text.

11. Fig. 5 caption: What is the time? Presumably these are monthly averages.

Authors: We clarified the caption.

12. P10, L16-17: This suggests you should be able to increase ensemble size using different days.

Authors: We agree with the reviewer's comment, assuming error variances remain fairly similar over time, that averaging over time provides a way to compensate for the lack of members. This conclusion may not apply if flux errors or transport errors vary during the averaging period.

13. P10, L21-22: "XCO2 variance spatial patterns (Fig. 6c) exhibit distinct maximum values located in the southwestern part of the domain," I don't see this. I see maxima (orange and red regions) in the eastern part of the domain. Also, the red region south of Lake Huron is similar to that seen in CO2 at 5 km in Fig. 6b. This is somewhat reassuring.

Authors: We corrected the description of the figure.

14. P14, L9-10: "One important point here is the calibration step performed before filtering. . ." The calibration steps in Diaz-Isaac et al. (2018a) and Appendix A were done over 18 June -21 July 2008 which encompasses the period of optimal filtering (Figs. 2-4). How applicable would this calibrated ensemble be to a different time period? My guess is that the calibration might yield a different selection of members for a different time period as the meteorology changes. So would you need to redo the calibration continually with time? Some addition discussion of the calibration process as a function of time might be useful here or in section 4.2.

Authors: We agree with the reviewer that calibration at different times may produce a different combination of model physics. For future studies, we would recommend a different approach where a predetermined set of members including both different

physics and random perturbations are used, without calibration, or with a limited number of members in the original ensemble. Otherwise, the calibration increases the computational time significantly. We discuss that point in section 4.2

15. P18, L5: "Figure 10" should be "Figure 12"

Authors: We corrected the reference.

16. Figure 12: Please add a legend to the bottom right panel of this figure, to label the curves. One can see which curve is which variable by comparing to the other panels, but a legend would be much more convenient for the reader.

Authors: We added a legend to the lower right panel.

17. P20, L4-6: As discussed in comment 1, the same qualifications need to be made here. Specifically, is this conclusion valid only in the context of additional temporal averaging?

Authors: We removed the 5-member ensemble considering that length scales were beyond our 750-km threshold for several days. We now recommend 8- to 10-member ensembles, and clarify the results for XCO2 which poses difficulties at the daily time scale.

Technical comments

1. P1, L4: "of which" should be "whose"

Authors: We corrected the sentence.

2. There are many instances of "fail at converging" which should be replaced by "fail to converge". i.e. P8L27, P8L30, P9L6, P9L15.

Authors: We replaced this terminology following the comments made by reviewer #1.

3. P10, L9: "dependence to" should be "dependence on"

Authors: We corrected the sentence.

4. P12, L13: "similar. . .than" should be "similar. . .to"

Authors: We corrected the sentence.

5. P13, L10: "difficulty" or "inability"?

Authors: We modified the sentence.

---

## Author Response (AR2)

Dear Chrisptoph Gerbig,

We have corrected the remaining occurences of "lack of convergence" that were forgotten in the revised manuscript. Except these requested modifications, the production files are identical to the discussion files.

Thank you for your time and for editing this paper.

Best regards,
Thomas Lauvaux, on behalf of the co-authors